# Cryo-EM structures of the human glutamine transporter SLC1A5 (ASCT2) in the outward-facing conformation

Xiaodi Yu[1†‡], Olga Plotnikova[1†], Paul D Bonin[1], Timothy A Subashi[1], Thomas J McLellan[1], Darren Dumlao[1], Ye Che[1], Yin Yao Dong[2§], Elisabeth P Carpenter[2], Graham M West[1], Xiayang Qiu[1], Jeffrey S Culp[1], Seungil Han[1]*

[1]Medicine Design, Pfizer Inc, Groton, United States; [2]Structural Genomics Consortium, University of Oxford, Oxford, United Kingdom

**Abstract** Alanine-serine-cysteine transporter 2 (ASCT2, SLC1A5) is the primary transporter of glutamine in cancer cells and regulates the mTORC1 signaling pathway. The SLC1A5 function involves finely tuned orchestration of two domain movements that include the substrate-binding transport domain and the scaffold domain. Here, we present cryo-EM structures of human SLC1A5 and its complex with the substrate, L-glutamine in an outward-facing conformation. These structures reveal insights into the conformation of the critical ECL2a loop which connects the two domains, thus allowing rigid body movement of the transport domain throughout the transport cycle. Furthermore, the structures provide new insights into substrate recognition, which involves conformational changes in the HP2 loop. A putative cholesterol binding site was observed near the domain interface in the outward-facing state. Comparison with the previously determined inward-facing structure of SCL1A5 provides a basis for a more integrated understanding of substrate recognition and transport mechanism in the SLC1 family.
DOI: https://doi.org/10.7554/eLife.48120.001

*For correspondence:
seungil.han@pfizer.com

[†]These authors contributed equally to this work

**Present address:** [‡]SMPS, Janssen Research and Development, Spring House, United States; [§]Neurosciences Group, Nuffield Department of Clinical Neuroscience, Weatherall Institute of Molecular Medicine, University of Oxford, Oxford, United Kingdom

## Introduction

SLC1A5 (ASCT2) catalyzes an obligatory $Na^+$-dependent antiport in which $Na^+$ together with an extracellular neutral amino acid are exchanged with an intracellular neutral amino acid (*Kanai and Hediger, 2004*; *Utsunomiya-Tate et al., 1996*). Among the solute carrier family 1 (SLC1) sub-family, it is the only one that is competent to transport glutamine, and it serves as the primary transporter of glutamine in cancer cells (*Fuchs and Bode, 2005*). Inhibition of SLC1A5 glutamine transport by small molecules and shRNA-mediated SLC1A5 knockdown have been shown to decrease endometrial cancer cell growth (*Marshall et al., 2017*). Similarly, SLC1A5 inhibition and knockdown suppressed mTORC1 signaling and induced rapid cell death in breast cancer cells (*van Geldermalsen et al., 2016*). In addition, SLC1A5 [-/-] mice are deficient in T cell receptor-mediated activation of the metabolic kinase mTORC1 and as a result exhibit improved clinical scores in experimental autoimmune animal models of multiple sclerosis and colitis (*Nakaya et al., 2014*).

The structure and function of SLC1 transporters have been studied using bacterial aspartate transporter homologues $Glt_{PH}$ from *Pyrococcus horikoshii* (*Yernool et al., 2004*; *Boudker et al., 2007*; *Reyes et al., 2009*; *Georgieva et al., 2013*; *Akyuz et al., 2015*; *Ji et al., 2016*), $Glt_{TK}$ from *Thermococcus kodakarensis* (*Jensen et al., 2013*; *Guskov et al., 2016*), human glutamate transporter SLC1A3 (*Canul-Tec et al., 2017*) and glutamine transporter SLC1A5 (*Garaeva et al., 2018*; *Garaeva et al., 2019*). All these transporters form a trimer of independently functioning protomers that uses an elevator mechanism to carry amino acids across membranes (*Akyuz et al., 2015*;

*Erkens et al., 2013*). Each protomer has a scaffold domain and a movable transport domain which slides and carries the solute. Glt$_{PH}$ structures have been reported in both the outward- and the inward-facing states (*Boudker et al., 2007*; *Reyes et al., 2009*; *Akyuz et al., 2015*; *Verdon et al., 2014*; *Verdon and Boudker, 2012*). Crystal structures of unbound and substrate-bound Glt$_{TK}$ have also been reported in the outward-facing conformation. The crystal structures of SLC1A3 were determined with competitive and allosteric inhibitors in the outward-facing conformation (*Canul-Tec et al., 2017*). Crystallographic studies of SLC1A3 required mutation of nearly 25% of its primary sequence suggesting the inherent technical challenges with this gene family (*Canul-Tec et al., 2017*). Recently, the cryo-EM structures of SLC1A5 in complex with glutamine or inhibitor were reported in the inward-facing state (*Garaeva et al., 2018*; *Garaeva et al., 2019*). To gain insights into structural features that permit substrate recognition in an outward-facing state, using an antibody fragment as a fiducial marker we have solved the cryo-EM structures of the unliganded transporter and its complex with glutamine at 3.5 and 3.8 Å resolution, respectively.

## Results

Our full-length SLC1A5 construct showed high expression and stability. (*Figure 1—figure supplement 1*). The presence of affinity tags used for purification did not affect the sodium-dependent glutamine uptake when HAP1 SLC1A5 knock-out cells were transiently transfected with full-length SLC1A5 but the observed transport activity was low, compared to the background. To assist structural determination, we generated its complex with a commercially available cKM4012 Fab fragment. It is reported that KM4012 antibody isolated through a cell-based screen inhibited glutamine-dependent cancer cell growth (*Suzuki et al., 2017*). The purpose of the Fab was to serve as a fiducial marker for particle alignment. Negative-stain electron microscopy revealed that one molecule of Fab was bound to each protomer of a SLC1A5 homotrimer (*Figure 1—figure supplement 1*). The three-fold symmetry was broken due to the dynamic nature of each Fab and protomer. Single-particle electron cryo-microscopy (cryo-EM) analysis without imposing any symmetry restraint resulted in a facile three-dimensional reconstruction at an overall resolution of 3.5 Å (Fourier shell correlation (FSC) = 0.143 criterion; *Figure 1—figure supplement 2*). The highly flexible Fab fragments were masked out to maximize the structural quality of SLC1A5 (*Figure 1—figure supplements 2,3*). The quality of the EM density map was sufficient to allow model building for residues 43–488 (*Figure 1—figure supplement 4*, *Table 1*). The limited resolution on the Fab region prevents a detailed and accurate analysis of the protein-fab interface.

The structure of SLC1A5 shows a homotrimer with an overall Glt$_{PH}$-like fold in an outward-facing conformation, Each SLC1A5 protomer contains two domains, a transport domain and a scaffold domain connected by the extracellular loop region 2 (ECL2a and ECL2b) (*Figure 1a-c*, *Figure 1—figure supplement 5*). The individual transport domain containing transmembrane helices TM3, TM6-TM8 and helical loops 1–2 (HP1-HP2) interacts exclusively with the scaffold domain from its own protomer. The scaffold domain including TM1-TM2 and TM4-TM5 is involved in inter-protomer interactions and forms a compact core. The ECL2 connects TM3 to TM5 via TM4 and bridges the two domains. ECL2a spans about 55 Å across a quarter of the transport domain, sequentially interacting with ECL3, TM8, TM7a, and HP2b on extracellular surface.

Superposition with the inward-facing state demonstrates that the scaffold domain remains relatively rigid, while the transport domain moves toward the cytoplasm and the substrate binding site shifts by 19 Å toward the cytoplasmic face (*Figure 2a-b*) (*Garaeva et al., 2018*). The ECL3 possessing one α-helix with ~1.5 turns connects the TM6 to the TM5 of the scaffold domain in the outward-facing state (*Figure 2a*). In the inward-facing state (PDB: 6GCT), this 1.5 turn α-helix in ECL3 merges into TM6 which extends the length of TM6 by about 9.5 Å (*Figure 2a,c*) (*Garaeva et al., 2018*). As a result, the shortened ECL3 further pulls the N-terminus of TM6 close to the TM5 causing the movement of TM6 toward the cytoplasm with ~40° tilt compared to the outward-facing state (*Figure 2a*). Simultaneously, to compensate the movement of the transport domain and the conformational changes of TM6, the N-terminus of TM3 unwinds by approximately 1.5 turns and extends the TM2-3 loop in the cytoplasmic space (*Figure 2b–c*). The extension of TM2-3 loop increases the distance between TM2 and TM3, resulting in a ~ 20° tilt compared to the outward-facing state (*Figure 2b*). The structurally symmetrical TM3 and TM6 serve as a platform which holds the core region of the transport domain (*Figure 2c*). During the transition between outward- and inward-facing states, the

**Table 1.** Data collection, reconstruction, and model refinement statistics

|  | SLC1A5_cKM4012 EMD-9187, PDB: 6MP6 | SLC1A5_cKM4012_L_Gln EMD-9188, PDB: 6MPB |
|---|---|---|
| **Data collection** | | |
| Microscope | Titan Krios | Titan Krios |
| Voltage (keV) | 300 | 300 |
| Nominal magnification | 22,500 x | 22,500 x |
| Exposure navigation | Stage Position | Stage Position |
| Electron exposure (e /Å$^2$) | 42 | 42 |
| Dose rate (e/pixel/sec) | 5 | 5 |
| Detector | K2 Summit | K2 Summit |
| Pixel size (Å)$^*$ | 0.543 | 0.543 |
| Defocus range (μm) | 1.2 to 2.5 | 1.2 to 2.5 |
| Micrographs Used | 3804 | 5228 |
| Final Refined particles (no.) | 165,067 | 253,220 |
| **Reconstruction** | | |
| Symmetry imposed | C1 | C1 |
| Resolution (global) | | |
| FSC 0.143 | 3.50 Å | 3.83 Å |
| Applied B-factor (Å$^2$) | −15 | −15 |
| **Refinement** | | |
| Protein residues | 1341 (SLC1A5) | 1344 (SLC1A5_L_Gln) |
| Map Correlation Coefficient | 0.815 | 0.785 |
| R.m.s deviations | | |
| Bond lengths (Å) | 0.006 | 0.005 |
| Bond angles (°) | 1.085 | 1.062 |
| Ramachandran | | |
| Outliers | 0.00% | 0.00% |
| Allowed | 7.88% | 8.93% |
| Favored | 92.12% | 91.07% |
| Poor rotamers (%) | 0.00% | 0.00% |
| MolProbity score | 1.62 | 1.67 |
| EMRinger score | 2.59 | 1.99 |
| Clashscore (all atoms) | 3.44 | 3.67 |

$^*$Calibrated pixel size at the detector

DOI: https://doi.org/10.7554/eLife.48120.002

changes in lengths of the TM3, TM2-3 loop, TM6 and ECL3 regions trigger the adjustment in the platform plane relative to the membrane plane, resulting in the movement of the transport domain coupled with a ~ 30° rotation (*Figure 2a-c*) (*Garaeva et al., 2018*). Notably, the well-defined ECL2a undergoes a large conformational rearrangement, swinging out by ~35° relative to the transport domain when compared with an inward-facing state of a cross-linked Glt$_{PH}$ structure (*Figure 2d*) (*Reyes et al., 2009*). In the outward-facing state, the ECL2a encompasses one side of the transport domain and is positioned to expose both the ECL4 and ECL5. However, in the inward-facing state, the ECL2a forms a sharp turn and sits on top of the transport domain covering a narrow groove formed by the ECL4 and ECL5 to accommodate rigid body movement of the transport domain (*Reyes et al., 2009*; *Verdon et al., 2014*; *Verdon and Boudker, 2012*; *Reyes et al., 2013*).

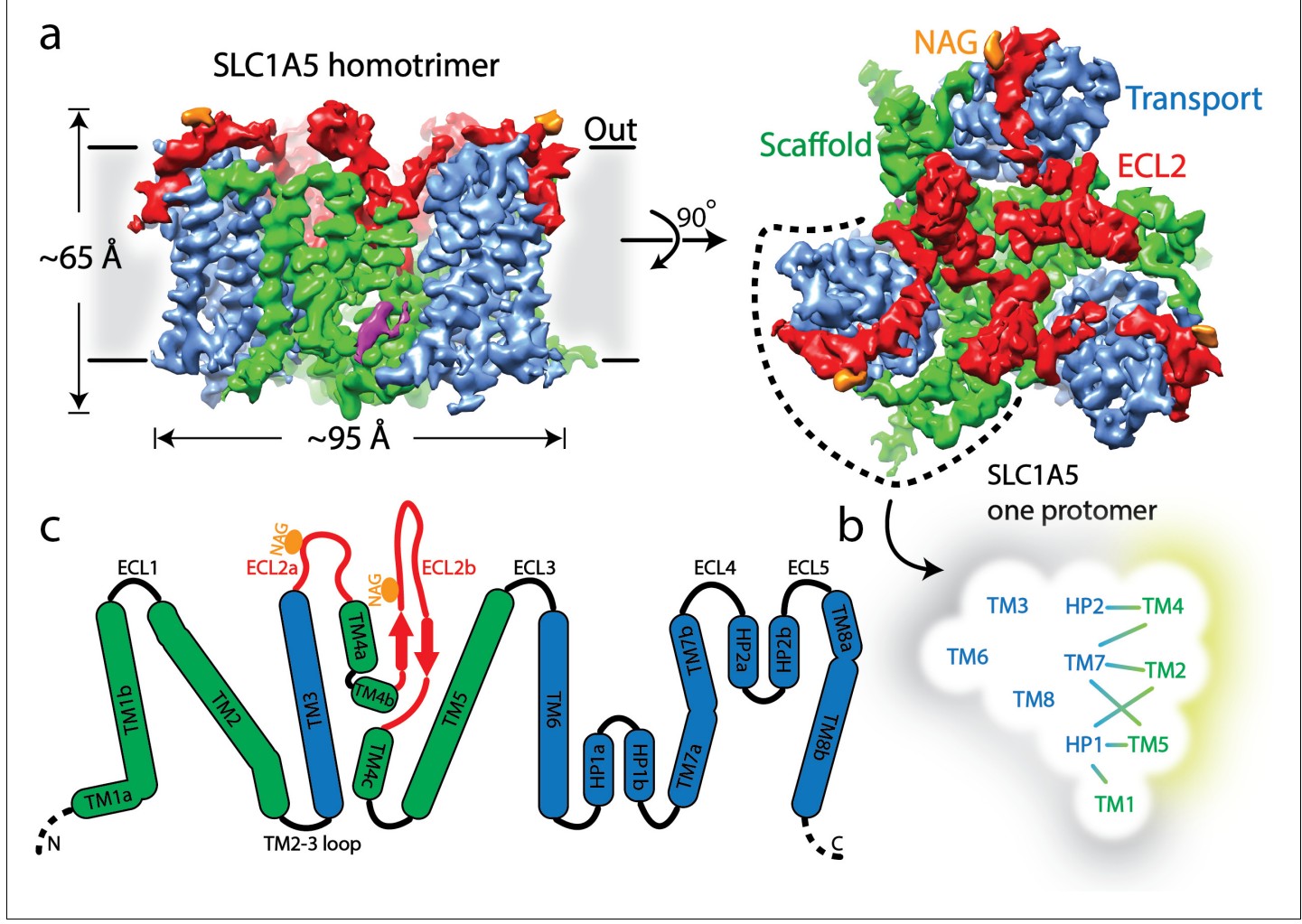

**Figure 1.** Cryo-EM structure of human SLC1A5. (**a**) A density map of SLC1A5 homotrimer viewed from the side of the membrane (left) and the extracellular face (right) highlighting the scaffold domain (green), transport domain (navy), ECL2 (red) and N-acetyl-D-glucosamine (NAG, in orange). The density assigned to CHS is colored in magenta. (**b**) Micelle/membrane and trimerization interfaces are highlighted in gray and wheat, respectively. Interactions between transport and scaffold domains are illustrated with lines. (**c**) A schematic representation of the domains present in human SLC1A5. The N-linked glycosylation sites are indicated by orange circle. Dashed lines represent residues disordered in the structure.

DOI: https://doi.org/10.7554/eLife.48120.003

The following figure supplements are available for figure 1:

**Figure supplement 1.** SLC1A5 protein purification, functional assays in cells and proteoliposomes and negative staining.
DOI: https://doi.org/10.7554/eLife.48120.004
**Figure supplement 2.** Cryo-EM analysis of SLC1A5-cKM4012 (Fab) complex.
DOI: https://doi.org/10.7554/eLife.48120.005
**Figure supplement 3.** Cryo-EM analysis of SLC1A5-cKM4012 (Fab) complex in the presence of L-glutamine.
DOI: https://doi.org/10.7554/eLife.48120.006
**Figure supplement 4.** Cryo-EM densities of the eight transmembrane helices with ECL loops of SLC1A5-cKM4012 (Fab).
DOI: https://doi.org/10.7554/eLife.48120.007
**Figure supplement 5.** Sequence alignment of human SLC1 transporters and two prokaryotic homologues.
DOI: https://doi.org/10.7554/eLife.48120.008

To understand the basis for substrate recognition by SLC1A5, we solved the cryo-EM structure of SLC1A5 in the presence of the substrate, L-Gln at 3.8 Å resolution (*Figure 1—figure supplement 3*). The density was sufficient to dock the L-Gln molecule (*Figure 3a*). The Gln binding pocket is formed by two oppositely oriented reentrant loops (HP1 and HP2), TM7 and TM8 of the transport domain (*Figure 3a–b*). The HP2 loop in the unliganded structure is in an open conformation providing a

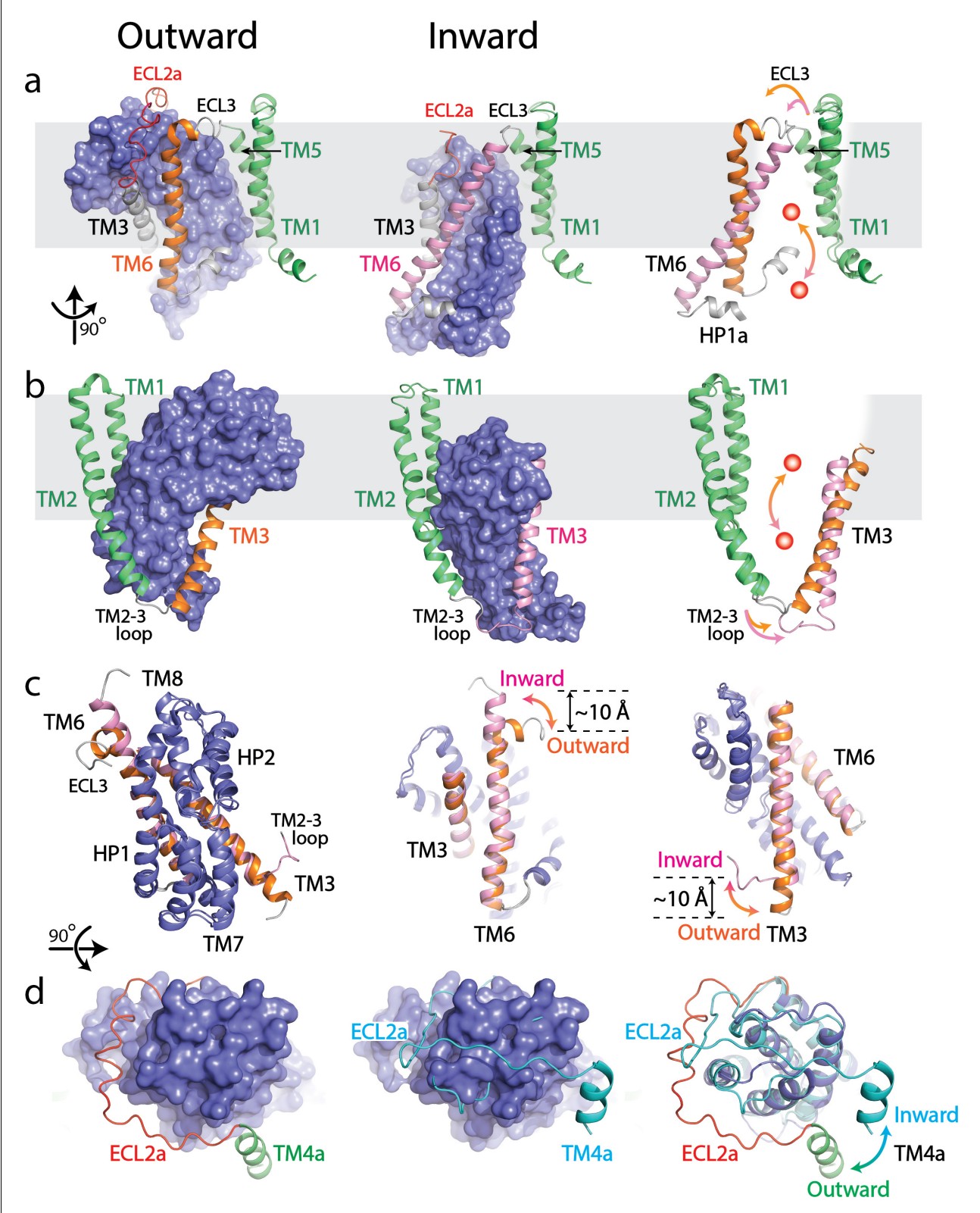

**Figure 2.** Structural comparison between outward- and inward-facing states of SLC1A5. (a and b) Structure of the SLC1A5 monomer viewed from the side of the membrane highlighting TM6 and TM3, respectively. Transport domains are represented as molecular surfaces and colored in navy. Scaffold domains are in green. TM6 and TM3 are colored in orange in the outward-facing state (left) and pink in the inward-facing state (middle, PDB: 6GCT). Overlay of scaffold domains of inward- and outward-facing states highlighting TM3 and TM6 (right). The glutamine substrate is shown as a red ball. (c)
*Figure 2 continued on next page*

*Figure 2 continued*

Superposition of the transport domains of SLC1A5 in the outward- and inward-facing (PDB: 6GCT) states (left). The conformational changes of TM6 (middle) and TM3 (right) between the outward-facing and inward-facing states are highlighted. (d) Transport domains of SLC1A5 in an outward-facing state (left) and a cross-linked Glt$_{PH}$ in inward-facing state (middle, PDB:3KBC) viewed from the extracellular face. The SLC1A5 transport domain in the outward-facing conformation serving as the reference is shown in molecular surface. The conformational changes of ECL2a between the outward- (in red) and inward-facing (in cyan) states are highlighted (right).
DOI: https://doi.org/10.7554/eLife.48120.009

direct access to the binding site. Upon Gln binding, HP2 loop acts as a gatekeeper and comes close to the serine-rich HP1 loop to shield the substrate from the extracellular surface (*Figure 3a*). Furthermore, the substrate binding interaction is maintained between the two states of SLC1A5 (*Figure 3c*) (*Garaeva et al., 2018*). Both α-amino and α-carboxylate groups of the substrate appear to be anchored by the carbonyl backbone of Ser351, the amide backbone and the sidechain of Ser353 in the HP1 loop and can be further stabilized by sidechains of Asp464 and Asn471 in TM8 (*Figure 3b*). Importantly, the Nε2 of the substrate, L-Gln, is within hydrogen bond distance to the sidechain of Asp464 in TM8. A smaller substrate, L-Asn, could fit into the binding site to satisfy the interaction with Asp464, consistent with the comparable K$_m$ values of human SLC1A5 for L-Asn and L-Gln determined in proteoliposomes (*Scalise et al., 2014*). The position of L-Gln in our SLC1A5 structure is similar to that of the L-Asp in the SLC1A3 crystal structure, suggesting a conserved-ligand binding pocket among SLC1 family members (*Figure 3d*) (*Yernool et al., 2004*; *Canul-Tec et al., 2017*).

Cys467, a unique residue among the SLC1 family, is at a suitable distance to donate its proton to the carbonyl group of the Gln substrate (*Figure 3b*, *Figure 1—figure supplement 5*) and could thus contribute to substrate selectivity. Mutation of Cys467 to Ala in SLC1A5 showed significant increase in K$_m$ for Gln (*Scalise et al., 2018*). The corresponding residue in all glutamate transporters of the SLC1 family is a conserved bulkier Arg residue (*Figure 1—figure supplement 5*). In SLC1A3, the Arg residue with its guanidinium group forms a hydrogen bond with the carboxylate group of the substrate Asp, and a favorable cation-π interaction with a Tyr residue (*Figure 3d*). As a result of the smaller amino acid at this location, SLC1A5 has an additional side cavity involving Phe393, Val436, Ala433, Val463 and Asp464, whereas no such cavity is present in SLC1A3 (*Figure 3b,d*). Besides the difference in the pocket size and shape, Arg in place of the Cys will lead to steric clashes with the Gln substrate in SLC1A5. The critical role of the Cys467 in SLC1A5 for substrate specificity was further supported by an equivalent mutation of threonine in SLC1A4 (Thr459) to a cysteine, resulting in alteration to L-Gln binding (*Scopelliti et al., 2018*) and inhibition by Cys modifying reagents in proteoliposome assays (*Pingitore et al., 2013*). Since the structure of SLC1A4 is unknown, we attempted to generate a model for SLC1A4 by replacing Cys with a threonine. Unfortunately, none of the three rotamers available for threonine can be satisfactorily accommodated because of steric clashes with backbone carbonyl atoms. This suggests that SLC1A4 must have a slightly different local conformation around the substrate binding site. Regardless of the local conformation that SLC1A4 adopts in this region, the side chain hydroxyl group of Thr459 (analogous to Cys467 in SLC1A5) is unlikely to form a productive hydrogen bonding interaction with the glutamine substrate and this could explain why the former transports only fewer amino acids. Ala390, a conserved residue in neutral amino acid transporters, SLC1A4 and SLC1A5, does not form any contact with the Gln substrate and could tolerate small neutral amino acid substrates (*Figure 3b*, *Figure 1—figure supplement 5*). In contrast, the structurally equivalent residue Thr in SLC1A3 forms a hydrogen bond with the carboxylate group of the substrate.

Stretches of residual EM densities were observed in a hydrophobic crevice near the domain interface in the presence or absence of glutamine (*Figure 1a*). Based on the shape of the density and the presence of cholesteryl hemisuccinate (CHS) during purification, we assigned this density to CHS. Using an unbiased lipidomics approach, CHS was detected as a major component in the sample used for cryo-EM. The identity of CHS was further confirmed by mass spectrometry using commercially available CHS as a reference (*Figure 3—figure supplement 1*).

The cryo-EM density supports a specific binding site for CHS in the outward-facing SLC1A5 conformation. Electron density suggests that CHS binds inside a hydrophobic pocket formed by TM3, TM4a, TM4c and TM7 from one protomer, as well as TM5 from the adjacent protomer (*Figure 3—figure supplements 2,3*). The putative lipid density was also observed in the inward-facing state of

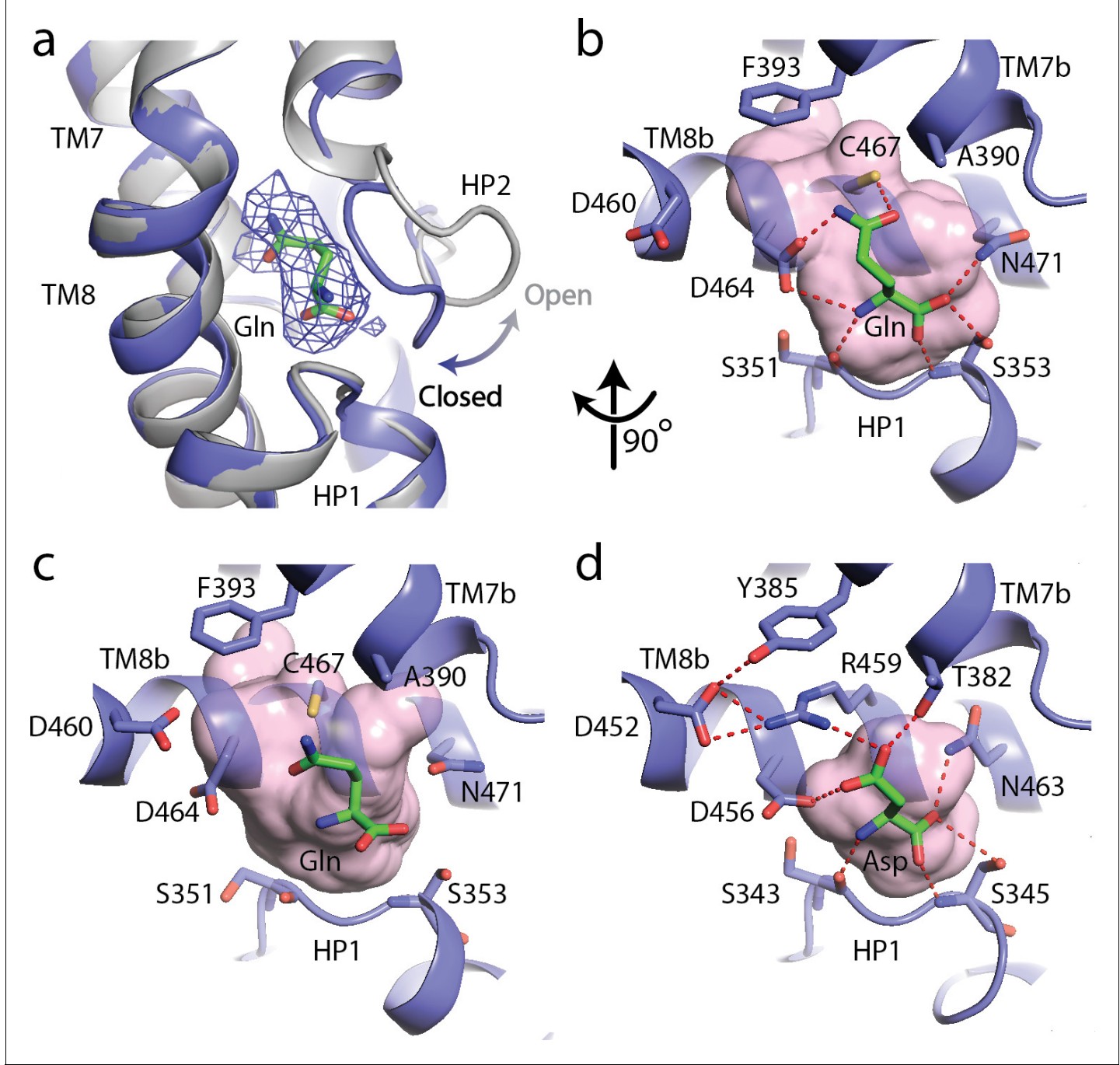

**Figure 3.** The substrate, L-glutamine-bound SLC1A5 structure. (a) Conformational change for the HP2 upon binding of L-glutamine in the outward-facing state of SLC1A5. The unliganded structure is shown in gray and the L-glutamine-bound structure is in navy. EM density over the ligand is shown as a blue mesh. (b and c) Zoomed in views of the substrate binding sites of the outward-facing and inward-facing states (PDB:6GCT) of SLC1A5, respectively. (d) Interaction of L-Asp in the outward-facing state of SLC1A3 (PDB code: 5LLU). Critical side chains are labeled (stick representation) with hydrogen bonds (red dashed lines). Molecular surface in the substrate binding sites are in pink.

DOI: https://doi.org/10.7554/eLife.48120.010

The following figure supplements are available for figure 3:

**Figure supplement 1.** The extracted ion chromatograph (above) and the mass spectrum (below) for CHS is shown in negative ion mode.

DOI: https://doi.org/10.7554/eLife.48120.011

**Figure supplement 2.** Potential CHS interaction in outward- and inward-facing conformation of SLC1A5.

DOI: https://doi.org/10.7554/eLife.48120.012

*Figure 3 continued on next page*

*Figure 3 continued*

**Figure supplement 3.** Cryo-EM densities of the substrate and putative CHS binding sites of SLC1A5, at the contour level 6σ.
DOI: https://doi.org/10.7554/eLife.48120.013

**Figure supplement 4.** Selective examples of allosteric druggable pockets at the lipid-exposed surface near the intracellular part of membrane proteins.
DOI: https://doi.org/10.7554/eLife.48120.014

SLC1A5, located in a hydrophobic pocket formed by TM3, TM4a, TM4c, and HP2a from one protomer, and TM5 from the adjacent protomer (*Garaeva et al., 2018*). Docking a CHS molecule in this putative density shows that the corresponding location is similar to that of the outward-facing state despite different pocket shape between two conformational states (*Figure 3—figure supplement 2*). Remarkably, the specific CHS-binding site in SLC1A5 overlaps with the UCPH$_{101}$ allosteric binding site reported in SLC1A3 structure (*Canul-Tec et al., 2017*) (*Figure 3—figure supplement 4*). The highly lipophilic UCPH$_{101}$ binds in the outward-facing conformation and blocks the movement of the transport domain relative to the scaffold domain. Additional putative lipid density was reported previously in the inward-facing state of SLC1A5, located between the transport and scaffold domains within the same protomer (*Garaeva et al., 2018*). The corresponding density was not observed in the outward-facing state of SLC1A5 due to differences in the local environment between the transport and scaffold domains.

## Discussion

Human SLC1A5 is the first eukaryotic sodium-dependent neutral amino acid transporter for which both the inward- and outward-facing conformational states have been mapped by cryo-EM. Our cryo-EM structures of unbound and substrate-bound SLC1A5 revealed that HP2 is the extracellular gate that controls entry of the substrate (*Boudker et al., 2007*; *Canul-Tec et al., 2017*). As a gatekeeper, HP2 in the closed conformation is near the serine-rich HP1, shielding the substrate from the extracellular surface (*Figure 3a*). The structure of the unliganded form of SLC1A5 provides an example for the HP2 loop in an open conformation and serves as a useful starting model for designing competitive inhibitors such as TBOA (*Boudker et al., 2007*; *Colas et al., 2015*). The opening of the HP2 loop increases the accessible surface area, exposing a previously unidentified cryptic pocket which could accommodate a large inhibitor that could specifically target the open conformation. In contrast, the closed conformation of HP2 has a small binding site, limiting the size of the inhibitors (*Colas et al., 2016*; *Singh et al., 2017*).

The availability of the three-dimensional structure of SLC1A5 in its outward-facing conformation open several paths forward for the development of drugs. The structure could be used for example identification of chemical leads by virtual screening methods. Secondly, generating additional structures with lead molecules based on substrate-analogs could provide a strong structure-based drug design platform for medicinal chemistry optimization of compound potencies. One important aspect of drug discovery is the need to engineer selectivity of lead compounds for the specific target in a given family. This is especially true for modulators of SLC1A5 where dialing out off-target activities against other closely-related sub-family members is highly desirable. Historically, this has been an area of strength for structure-based design. Structures have also been incredibly useful for optimization of ADME properties of compounds which is a critical element in the overall drug discovery process. Lastly, we believe the most relevant conformation to target for SLC1A5 inhibitors is the outward-facing conformation that we have described in our current work as that might be readily accessible for ligands from extracellular space.

The human SLC1A5 represents a distinct SLC1 subfamily identifiable in part by the Cys and Ala residues for substrate specificity and the unique conformation of the two ECL2a and ECL2b loops allowing for structural flexibility at TM4b for the key trimer interaction. Furthermore, ECL2a encircles the transport domain in the outward-facing state, covering a large part of the interaction surface for the inward facing state. In the inward-facing state the ECL2a is repositioned, exposing the interaction surface for the scaffold domain (*Figure 4*) (*Garaeva et al., 2018*).

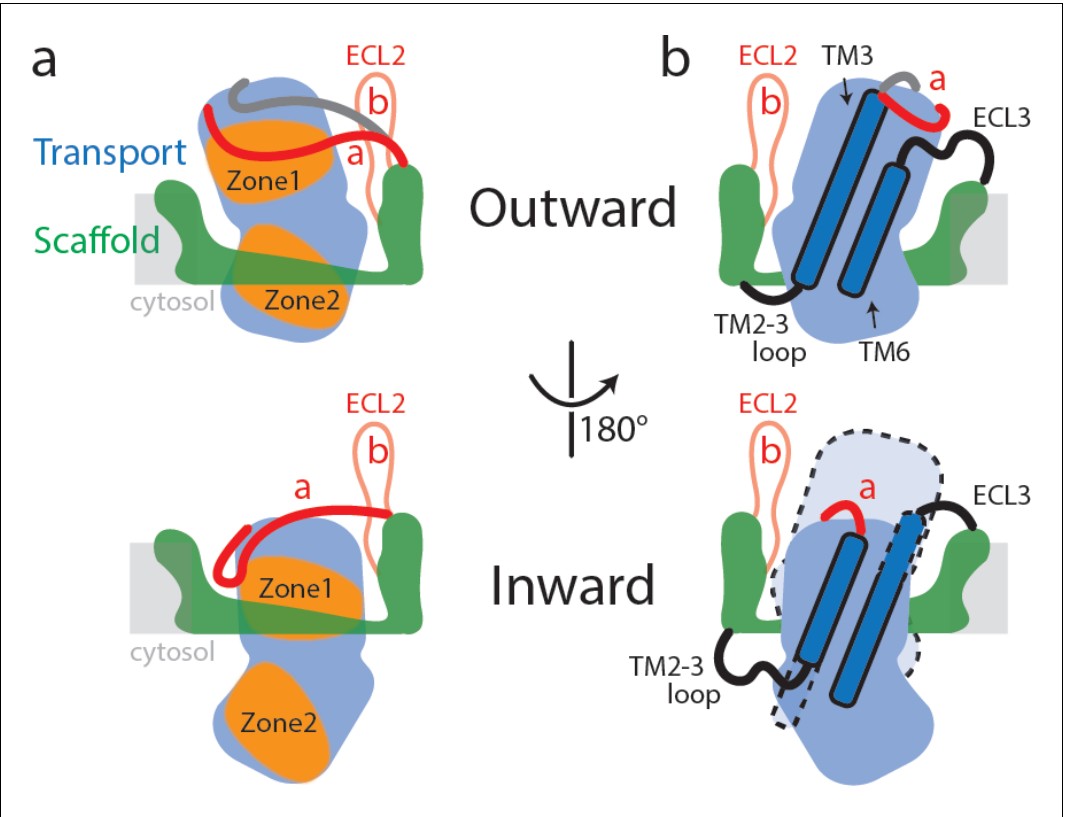

**Figure 4.** Schematic representation of conformational changes in SLC1A5 during substrate transport. (a) In the outward-facing state (top), the Zone two from the transport domain interacts with the scaffold domain, with the Zone one partially occupied by the ECL2a. Two different poses of the ECL2a were observed (shown in red and dark gray lines): one is situated at the side of the transport domain (red line); the other is crossing over the crests of ECL4 and ECL5 (dark gray line). In the inward-facing state (bottom), the ECL2a has to be repositioned, releasing the Zone one to interact with the scaffold domain. Residues from Zone one and Zone two are highlighted (*Figure 1—figure supplement 5*). (b) The inherent plasticity around the TM2-3 loop, ECL3, TM3, and TM6 regions (highlighted in dash lines) allows the transitions between the outward-facing (top) and inward-facing states (bottom). The relative position of transport domain in the outward-facing state is highlighted in dash (bottom panel in b). The regions of SLC1A5 are colored based on the *Figure 1a*. The light gray line depicts membrane boundary.

DOI: https://doi.org/10.7554/eLife.48120.015

The existence of a potential allosteric binding pocket observed in the outward- and inward-facing SLC1A5 structures could facilitate the design of selective inhibitors that can bind to a site that is remote from the conserved substrate-binding site. Such allosteric druggable pockets at the lipid-exposed surface near the intracellular portions of membrane proteins have recently been structurally elucidated in several proteins and this raises new opportunities for drug design (*Figure 3—figure supplement 4*). We believe that these findings provide new and exciting structural insights and opportunities for antagonizing glutamine metabolism at the transporter level. Our structures of human SLC1A5, in both unbound and substrate-bound forms, represent the first high resolution cryo-EM structures of SLC1 family captured in the outward-facing state. These structures are an early example of the value that cryo-EM may bring to structure-based drug design.

# Materials and methods

**Key resources table**

| Reagent type (species) or resource | Designation | Source or reference | Identifiers | Additional information |
|---|---|---|---|---|
| Antibody | Mouse monoclonal Anti-SLC1A5 | Creative Biolabs Inc | TAB-1010CLV | 1 ml, 1 mg/ml |
| Strain, strain background (*Homo sapiens*) | HEK293S GnTI⁻ cells | ATCC | CRL-3022 RRID: CVCL_A785 | |
| Recombinant DNA reagent | pcDNA3.1⁺ | Life Technologies | V79020 | |
| Chemical compound, drug | Opti-MEM reduced serum media | Life Technologies | 31985062 | |
| Strain, strain background (*Homo sapiens*) | HAP1 SLC1A5 knock out cells | Horizon Discovery | HZGHC005452c002 | |
| Chemical compound, drug | Iscove's modified Dulbecco's medium | Gibco | 12440 | |
| Chemical compound, drug | Cell dissociation buffer | Gibco | 13151 | |
| Chemical compound, drug | Fetal bovine serum | Gibco | 16000044 | |
| Software, algorithm | MotionCorr2 1.1.0 | *Zheng et al., 2017* | http://msg.ucsf.edu/em/software/motioncor2.html | |
| Software, algorithm | Relion v 2.0 | *Kimanius et al., 2016* | https://www2.mrc-lmb.cam.ac.uk/relion/ | |
| Software, algorithm | Phenix 1.14 | *Adams et al., 2010* | http://phenix-online.org/ | |
| Software, algorithm | Coot 0.8.9.1 | *Emsley and Cowtan, 2004* | https://www2.mrc-lmb.cam.ac.uk/personal/pemsley/coot/ | |
| Software, algorithm | Pymol 2.0 | Schrodinger LLC | https://pymol.org/2/ | |
| Software, algorithm | Chimera 1.12 | *Pettersen et al., 2004* | https://www.cgl.ucsf.edu/chimera/ | |
| Software, algorithm | Gctf | *Zhang, 2016* | https://www.mrc-lmb.cam.ac.uk/kzhang/ | |
| Software, algorithm | cisTem | *Grant et al., 2018* | https://cistem.org/ | |

## Protein expression

The gene encoding SLC1A5 was cloned into pcDNA3.1⁺ (Life Technologies) where it was fused to a C-terminal His10-Flag affinity double tag separated by TEV protease cleavage site. The protein was expressed in HEK293S GnTI⁻ cells (ATCC CRL-3022, RRID: CVCL_A785) grown in Expi293 Expression Media (Life Technologies) to densities $2.5 \times 10^6$ cells ml$^{-1}$. Cells were transiently transfected in Opti-MEM reduced serum media (Life Technologies) using ExpiFectamine Transfection Kit (Life Technologies). Cells were collected at 48 hr after transfection. Although commercially available, MAb cKM4012 (Creative Biolabs Inc) was cloned, expressed and purified in-house to meet reagent quantity requirements (*Suzuki et al., 2017*; *Shiraishi et al., 2012*).

## SLC1A5 and SLC1A5_cKM4012 (Fab) complex purification

Cells were resuspended at 1:10 (w/v) ratio in 25 mM HEPES pH 7.0, 300 mM NaCl, 0.2 mM Tris (2-carboxyethyl) phosphine (TCEP), buffer supplemented with EDTA free protease inhibitor cocktail (Roche) and 2.5 μl/ml of Benzonase nuclease (Sigma), and disrupted in Microfluidizer processor M110L (Microfluidics) at 12,000 psi. Cell lysate was clarified by centrifugation at 4,000 g for 30 min, and membrane fraction was collected by ultracentrifugation at 225,000 g for 60 min. Membrane pellets were homogenized at 1:5 (w/v) ratio in 25 mM HEPES pH 7.0, 300 mM NaCl, 1% lauryl maltose neopentyl glycol (LMNG) (Anatrace), 0.1% cholesterol hemisuccinate (CHS) (Anatrace), 0.2 mM TCEP

buffer followed by 2 hr solubilization with gentle agitation at 4°C. Insoluble material was removed by ultracentrifugation at 290,000 g for 60 min. Solubilized material was incubated overnight at 4°C with anti-flag M2 affinity agarose resin (Sigma) with gentle agitation. 1 ml of the resin efficiently captured solubilized SLC1A5 from 10 g of cells. Unbound material was removed by centrifugation at 1,000 g for 10 min and the resin was washed with 25 mM HEPES pH 7.0, 300 mM NaCl, 0.003% LMNG, 0.2 mM TCEP buffer (buffer A) three times. SLC1A5 was eluted with buffer A supplemented with 0.2 mg/ml Flag peptide (CPC Scientific) and concentrated to 1 mg ml$^{-1}$ using Amicon Ultra Ultracell-100K (Millipore Sigma) centrifugal filter. The affinity purified protein was applied to Superose 6 Increase 10/300 gel filtration column (GE Healthcare) equilibrated with buffer A. All protein purification steps performed on ice or at 4°C at all times.

Fab fragments of cKM4012 monoclonal antibodies were generated using the Fab Preparation Kit (Pierce) and further purified on Superdex 75 column equilibrated with 30 mM HEPES pH 7.0, 100 mM NaCl, 0.003% LMNG, 10 mM L-glutamine (L-Gln). The same buffer without L-Gln was used for samples prepared in the absence of glutamine. When glutamine was present, SLC1A5/cKM4012 (Fab)/L-Gln complex was formed in the followed order: 0.85 ml of SLC1A5 at 16 μM was mixed with 0.05 ml of L-Gln at 200 mM to achieve 10 mM final L-Gln concentration, incubated 20 min in ice, mixed with 1 ml of cKM4012 Fab fragments at 16 μM, incubated 20 min in ice, concentrated to 0.1 ml on Amicon Ultra Ultracell-100K (Millipore Sigma) centrifugal filter. The complex was loaded to Superose 6 10/300 Increase filtration column (GE Healthcare) equilibrated with 25 mM HEPES pH 7.0, 100 mM NaCl, 0.003% LMNG and either 0 mM or 10 mM L-Gln. Two top peak fractions were taken and concentrated to 2 mg/ml.

## L-Glutamine uptake assay

HAP1 SLC1A5 knock out (SLC1A5KO) cells (Horizon Discovery HZGHC005452c002) contain a 1 bp insertion in a coding exon of SLC1A5 and do not express SLC1A5 (as described by the vendor and confirmed by us). SLC1A5KO cells were transiently transfected by static electroporation using a Max-Cyte STX system. Viable cells ($10^6$ per mL) were transfected with 1.0 μg of pcDNA3.1 (empty vector) or with 1 μg of either pcDNA3.1-FL-SLC1A5 or pcDNA3.1-FL-SLC1A5-TEV-His-Flag using the OC-100 or OC-400 processing assembly (electroporation cell) and conditions specified by the vendor. Transfected cells were then cultured at 37°C in a humidified environment in 5% carbon dioxide. After 24 hr, transfected cells were removed flasks and frozen for later use. Glutamine uptake was determined by measuring the incorporation of radio-labeled glutamine into SLC1A5KO cells transfected with the indicated DNA. Specifically, frozen SLC1A5KO cells were thawed then cultured overnight in T-175 flasks in Iscove's modified Dulbecco's medium (Gibco 12440) supplemented with 10% fetal bovine serum. At the time of assay, the cells were removed from flasks with cell dissociation buffer (Gibco 13151), centrifuged at 800 x g, re-suspended in assay buffer (25 mM Tris pH 7.4, 2 mM KCl, 1 mM MgCl2, 1 mM CaCl2, 5 mM glucose with or without and 100 mM NaCl) and plated at 50,000 cells per well in a 384-well Cytostar-T scintillation microplate (PerkinElmer RPNQ0166). The plate was then incubated at 37°C in a humidified environment in 5% carbon dioxide. After ~60 min, the plate was removed from the incubator and placed at room temperature (RT). After ~15 min, the assay was initiated by the addition of a mixture of unlabeled glutamine and $^{14}$C-labeled glutamine (PerkinElmer NEC4150) prepared in assay buffer. The plate was then covered with a plastic seal and [$^{14}$C]-glutamine uptake was determined at 40 min by scintillation counting with a MicroBeta2 (2450 Microplate Counter; Perkin Elmer). The final assay volume was 40 μL and the final assay conditions were 25 mM TRIS pH 7.4, 2 mM KCl, 1 mM MgCl$_2$, 1 mM CaCl$_2$, 5 mM glucose, with or without 100 mM NaCl and 30 μM glutamine/[$^{14}$C]-glutamine (50 dpm/pmol).

## Grid preparation and data acquisition

3.5 μL of 2.0 mg/ml purified SLC1A5_cKM4012 (Fab) with/without L-Gln complex was applied to the glow-discharged Quantifoil Au R1.2/1.3 grid (Structure Probe), and subsequently vitrified using a Vitrobot Mark IV (FEI Company). In order to overcome an orientation bias, n-octyl-β -d-glucopyrano-side (BOG, Anatrace) was added to the sample prior freezing. Cryo grids were loaded into a Titan Krios transmission electron microscope (ThermoFisher Scientific) operating at 300 keV with a Gatan K2 Summit direct electron detector. Images were recorded with SerialEM in super-resolution mode with a super resolution pixel size of 0.543 Å and a defocus range of 1.2 to 2.5 μm. Data were

collected with a dose rate of 5 electrons per physical pixel per second, and images were recorded with a 10 s exposure and 250 ms subframes (40 total frames) corresponding to a total dose of 42 electrons per Å$^2$. All details corresponding to individual datasets are summarized in *Table 1*.

## Electron microscopy data processing

For SLC1A5-cKM4012 (Fab) complex, a total of 3804 dose-fractioned movies were gain-corrected, 2 x binned (resulting in a pixel size of 1.086 Å), and beam-induced motion correction using Motion-Cor2 (*Zheng et al., 2017*) with the dose-weighting option. The SLC1A5_cKM4012 (Fab) particles were automatically picked from the dose-weighted, motion corrected average images using Gautomatch. CTF parameters were determined by Gctf (*Zhang, 2016*). A total of 743,525 particles were then extracted using Relion 2.0 (*Scheres and Chen, 2012*) with a box size of 224 pixels. The 2D, 3D classification and refinement were performed with Relion 2.0. Two rounds of 2D classification and one round of 3D classification were performed to select the homogenous particles. 3D classification results showed only the outward-facing state with the substrate binding site accessible from the extracellular side. After selecting particle coordinates, per-particle CTF estimation was refined using the program Gctf (*Zhang, 2016*). One set of 170,751 particles was then submitted to 3D auto-refinement. All 3D classifications and 3D refinements were started from a 60 Å low-pass filtered version of an ab initio map generated with VIPER (*Penczek et al., 1994*). The particles were re-centered using the refined particle offsets before being re-extracted, followed by parameter conversion and final Auto_Refine in cisTEM (*Grant et al., 2018*). Auto_Refine in cisTEM, using a soft binary mask that removed the micelle and Fab regions, yielded the final reconstruction at 3.5 Å global resolution, improving the resolution by ~0.3 Å. The global resolution was evaluated using conventional Fourier Shell Correlation analysis. Prior to visualization, all density maps were sharpened by applying different negative temperature factors (−15,–30, and −50 Å$^2$) using automated procedures (*Rosenthal and Henderson, 2003*), along with the half maps, were used for model building. Local resolution was determined using ResMap (*Kucukelbir et al., 2014*) (*Figure 1—figure supplement 2*). Data collection, processing and refinement of SLC1A5-cKM4012 (Fab) with L-Gln substrate complex were performed similarly as described for SLC1A5-cKM4012 (Fab) (*Figure 1—figure supplement 3*). Fab signals subtraction followed by refinements with/without C3 symmetry operation did not improve the resolutions for both the SLC1A5-cKM4012 (Fab) and SLC1A5-cKM4012 (Fab)_Gln complexes. The reason could be the Fab-bound SLC1A5 is perhaps pseudo-3 fold symmetric. The presence of the Fab disrupts the low-resolution pseudo-symmetry and facilitates particle alignment. However, the flexibility of the Fabs harms the high-resolution reconstruction at the SLC1A5 trimer regions. A soft binary mask was applied to the SLC1A5 trimer region during the last several rounds of Auto_Refine in cisTEM to improve the local resolution.

## Model building and refinement

The initial template of the human SLC1A5 was derived from a homology-based model calculated by SWISS-MODEL (*Arnold et al., 2006*). The model was docked into the EM density map using Chimera (*Pettersen et al., 2004*) and followed by manually adjustment using COOT (*Emsley and Cowtan, 2004*). For the SLC1A5_L-Gln modeling, one L-glutamine molecular was generated and manually docked into the density using COOT. Each model was independently subjected to global refinement and minimization in real space using the module phenix.real_space_refine in PHENIX (*Adams et al., 2010*) against separate EM half-maps with default parameters. The model was refined into a working half-map, and improvement of the model was monitored using the free half map. The geometry parameters of the final models were validated in Coot and using MolProbity and EMRinger (*Barad et al., 2015*). These refinements were performed iteratively until no further improvements were observed. The final refinement statistics were provided in *Table 1*. Model overfitting was evaluated through its refinement against one cryo-EM half map. FSC curves were calculated between the resulting model and the working half map as well as between the resulting model and the free half and full maps for cross-validation (*Figure 1—figure supplement 4*). Figures were produced using PyMOL (*Schrodinger LLC, 2015*) and Chimera (*Pettersen et al., 2004*).

## SLC1A5 proteoliposomes production

Liposomes were prepared according to Pingitore et al. with modifications (*Pingitore et al., 2013*). A 9:1 mixture of L-α-phosphatidylcholine (PC) (Sigma) and 18:1 biotinyl cap phosphoethanolamine (PE) (Avanti) was dried under $N_2$ gas stream, rehydrated in water by vigorous mixing on a vortex mixer followed by extrusion through 400 nm polycarbonate membranes (Sigma) resulting in 6.7% liposomes. SLC1A5 proteoliposomes were formed during detergent-mediated reconstitution. First, pre-made liposomes were mixed with C12E8 to final 1.4% of liposomes and 1.7% of C12E8 in 20 mM Tris, pH 7.0, 10 mM L-Gln followed by 30 min incubation at RT. Second, purified SLC1A5 in 20 mM Tris, pH 7.0, 100 mM 100 NaCl, 10 mM L-Gln, 0.05% n-dodecyl-β-D-maltopyranoside (DDM) and 6 mM β- mercaptoethanol was added to lipid-detergent-micellar solution to make 2000:1 lipid to protein ratio. The protein-lipid-detergent mixture was incubated 30 min at RT before detergents were removed through absorption on hydrophobic resin Amberlite XAD-4. To exchange to an uptake buffer, proteoliposomes were passed through the 7K MWCO Zeba spin desalting columns (Thermo-Fisher Scientific) equilibrated with 20 mM Tris, pH 7.0, 15 mM Sucrose to balance internal osmolality.

## L-glutamine uptake assay using proteoliposomes

L-glutamine uptake was determined by measuring the incorporation of radio-labeled L-glutamine into biotinylated-SLC1A5 proteoliposomes. Specifically, the assay was initiated by the addition of biotinylated-SLC1A5 proteoliposomes into triplicate wells of a 96-well polypropylene plate with each well containing the indicated concentration of NaCl and 50 µM L-glutamine/L-[3,4-[3]H)]-glutamine (100 dpm/pmol). After the indicated time at room temperature (~22°C), the assay was stopped by the addition of 15 µl of ice-cold streptavidin beads (Pierce Streptavidin Agarose) prepared in 20 mM Tris, pH 7.0, 15 mM sucrose buffer. After 10 min at room temperature the biotin proteoliposome/ streptavidin bead complex was transferred to a 96-well GF/C filter plate (PerkinElmer) plate (pre-blocked with 0.1% BSA and washed x 2 in 20 mM Tris, pH 7.0, 15 mM sucrose buffer). The plate was then vacuum filtered and then washed three times with 500 µl per well of ice-cold 20 mM Tris, pH 7.0, 15 mM sucrose buffer. The plate was then air dried overnight and 50 ul of Betaplate Scint (PerkinElmer) was added to each well of the 96-well plate. The plate was then sealed and glutamine uptake was determined by scintillation counting with a MicroBeta2 (PerkinElmer).

## Cholesteryl hemisuccinate (CHS) detection by untargeted LC/MS

Purified SLC1A5 protein sample was treated with 10X acetonitrile for 1 hr to denature proteins. Precipitated proteins were pelleted using centrifugation at 15000 x g for 10 min and the supernatant was transferred to a new vial. The supernatant was dried completely using a speed-vac and resuspended with 100 µl of 2:2:1 isopropanol:acetonitrile:water. CHS was detected using high resolution mass spectrometer (Q-Exactive plus) in series with an Agilent 1290 UPLC pump. We modified the waters application note: Lipid Separation using UPLC with Charged Surface Hybrid Technology. Briefly, the mobile phase system consisted of (A) acetonitrile: water (60:40) with 10 mM ammonium formate and 0.1% formic acid and (B) isopropanol:acetonitrile (90:10) with 10 mM ammonium formate and 0.1% formic acid. Metabolites were separated using an Acquity UPLC CSH C18 2.1 × 100 mm, 1.7 µm column heat to 55°C. 10 µL of each sample were subjected to the following gradient: time 0 = 40% B, time 2 = 43% B, time 2.1 = 50% B, time 12 = 54% B, time 12.1 = 70% B, time 18 = 99% B, time 18.1 = 40% B, time 20 = 40% B. The mass spectrometer was set with the following parameters: mass range = 100–1500 m/z, negative mode, 70,000 resolution, AGC target = 1e6, Maximum IT = 100 ms, Centroid, ddMS2 resolution = 30,000, AGC target 1e5, Maximum IT 50 ms, Top N = 5, Loop count = 5, Isolation window = 4.0 m/z, CE = 30. The data were processed using Compound Discoverer 2.0.

## Acknowledgements

We thank James Smith, Chia-Chin Lee, Elizabeth Dushin and Kelsey Oleynek for assisting with cloning, expression and protein purification of SLC1A5. We also thank Mark Tibbitts for cell transfection. We thank Kingsley Sumner for IT support and Ravi Kurumbail for critical reading.

## Additional information

### Competing interests

Xiaodi Yu, Olga Plotnikova, Paul D Bonin, Timothy A Subashi, Thomas J McLellan, Darren Dumlao, Ye Che, Graham M West, Xiayang Qiu, Jeffrey S Culp, Seungil Han: is affiliated with Pfizer Inc. The author has no other competing interests to declare. The other authors declare that no competing interests exist.

### Funding

YD and EPC are members of the SGC, (Charity ref: 1097737) funded by AbbVie, Bayer Pharma AG, Boehringer Ingelheim, the Canada Foundation for Innovation, Genome Canada, GlaxoSmithKline, Janssen, Lilly Canada, Merck & Co., Novartis, the Ontario Ministry of Economic Development and Innovation, Pfizer, São Paulo Research Foundation-FAPESP and Takeda, as well as the Innovative Medicines Initiative Joint Undertaking ULTRA-DD grant 115766 and the Wellcome Trust 106169/Z/14/Z. No additional external funding was received for this work.

### Author contributions

Xiaodi Yu, Conceptualization, Data curation, Formal analysis, Supervision, Validation, Investigation, Visualization, Methodology, Writing—original draft, Project administration, Writing—review and editing; Olga Plotnikova, Resources, Data curation, Formal analysis, Validation, Visualization, Methodology, Writing—original draft; Paul D Bonin, Resources, Data curation, Formal analysis, Validation, Methodology, Writing—original draft; Timothy A Subashi, Resources, Data curation, Formal analysis, Methodology; Thomas J McLellan, Elisabeth P Carpenter, Resources, Methodology; Darren Dumlao, Resources, Validation, Methodology; Ye Che, Yin Yao Dong, Resources, Formal analysis, Validation, Methodology; Graham M West, Resources, Formal analysis, Methodology, Writing—original draft; Xiayang Qiu, Conceptualization, Resources, Formal analysis, Validation, Methodology, Writing—original draft; Jeffrey S Culp, Conceptualization, Supervision, Validation, Investigation, Methodology, Writing—original draft; Seungil Han, Conceptualization, Data curation, Supervision, Validation, Investigation, Visualization, Methodology, Writing—original draft, Project administration, Writing—review and editing

### Author ORCIDs

Seungil Han (iD) https://orcid.org/0000-0002-1070-3880

### Decision letter and Author response

Decision letter https://doi.org/10.7554/eLife.48120.026
Author response https://doi.org/10.7554/eLife.48120.027

## Additional files

### Supplementary files

• Transparent reporting form DOI: https://doi.org/10.7554/eLife.48120.016

### Data availability

All the cryo-EM data were deposited to the Protein Data Bank (PDB ID: 6MP6, 6MPB) and the EMDB (EMD-9187, EMD-9188) for immediate release upon publication.

The following datasets were generated:

| Author(s) | Year | Dataset title | Dataset URL | Database and Identifier |
|---|---|---|---|---|
| Yu X, Plotnikova O, Bonin PD, Subashi TA, McLellan TJ, Dumlao D, Che Y, Dong YY, Carpenter EP, West GM, | 2019 | SLC1A5_cKM4012 | https://www.rcsb.org/structure/6MP6 | Protein Data Bank, 6MP6 |

| | | | | | |
|---|---|---|---|---|---|
| Qiu X, Culp JS, Han S | | | | | |
| Yu X, Plotnikova O, Bonin PD, Subashi TA, McLellan TJ, Dumlao D, Che Y, Dong YY, Carpenter EP, West GM, Qiu X, Culp JS, Han S | 2019 | SLC1A5_cKM4012_L_Gln | https://www.rcsb.org/structure/6MPB | Protein Data Bank, 6MPB |
| Yu X, Plotnikova O, Bonin PD, Subashi TA, McLellan TJ, Dumlao D, Che Y, Dong YY, Carpenter EP, West GM, Qiu X, Culp JS, Han S | 2019 | SLC1A5_cKM4012 | https://www.ebi.ac.uk/pdbe/entry/emdb/EMD-9187 | Electron Microscopy Data Bank, EMD-9187 |
| Yu X, Plotnikova O, Bonin PD, Subashi TA, McLellan TJ, Dumlao D, Che Y, Dong YY, Carpenter EP, West GM, Qiu X, Culp JS, Han S | 2019 | SLC1A5_cKM4012_L_Gln | https://www.ebi.ac.uk/pdbe/entry/emdb/EMD-9188 | Electron Microscopy Data Bank, EMD-9188 |

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
