## [Decision Letter]

Thank you for submitting your article "Structural basis for the transport mechanism of the human glutamine transporter SLC1A5 (ASCT2)" for consideration by *eLife*. Your article has been reviewed by Olga Boudker as the Senior Editor, a Reviewing Editor and three peer reviewers. The following individuals involved in review of your submission have agreed to reveal their identity: Robert John Vandenberg (Reviewer #3); Joel Meyerson (Reviewer #4).

The reviewers have discussed the reviews with one another and the Reviewing Editor has drafted this decision to help you prepare a revised submission.

Summary:

The neutral amino acid transporter ASCT2 is a member of the SLC1 family of transporters and a target of cancer therapeutics. The manuscript by Yu et al. presents the determination of a structure of ASCT2 in an outward-facing conformation, complementing a previously published structure of an inward facing structure of ASCT2, and providing details of the substrate binding site, which should prove invaluable for the design of drugs to inhibit the transporter.

The reviewers appreciate the high quality of the structural work and general importance of the structure, but note that it requires a more direct connection to physiology, e.g. the differences between exchangers and transporters, and/or to drug design. To focus the framing of the article we therefore strongly suggest that you rewrite the article as a Short Report, focus on the substrate and putative CHS binding sites, and exclude the HDX data as explained below.

Essential revisions:

The following are the points that should be considered when improving the physiological context of the structure in the revised manuscript:

1) Based on the structures, is it clear why ASCT2 and ASCT1 have different substrate specificity?

2) From the drug design perspective please provide more specific comments on the potential applications of the structure for the development of drugs.

3) It is not clear whether the protein imaged without L-Gln was in high Na^+^ concentration. If yes, then the structure likely represents a Na^+^-bound transporter, which should be evident from the arrangement of the Na^+^-binding residues. If not, or if the affinity for Na^+^ is low, then one would expect to find disrupted Na^+^ binding sites, which again should be visible in the structure.

4) ASCTs are obligate Na^+^-dependent exchangers, while EAATs are concentrative pumps. Could it be rationalized from the structures, why ASCT2 cannot return to the outward-facing states without the bound amino acid and Na^+^ ions?

5) Mutations have been made in GltPh mimicking ASCT2 (including R to C mutation), but when the structure was solved with Gln, the HP2 loop was still open. Can it be rationalized from the current structure what is the difference between the binding pockets and/or HP2 between ACST2 and GltPh that allows larger substrates (even when R is mutated to C)?

6) The manuscript methodically indexes many structural features including protomer interactions, helix orientations, key amino acid positions, glycosylation sites, glutamine and CHS binding sites, etc. However, this style is primarily descriptive, so if the authors intend to retain these details, they should to be enhanced with a clearer narrative explaining the significance of the structural findings. Below are three examples.

The authors point out that "based on sequence alignment, both SLC1A4 and SLC1A5 have the longest ECL2a region compared to other SLC1 family members including prokaryotic homologues." What is the significance of this?

Similarly, the authors write: "Importantly, the ECL2b was reported to play a critical role in determining the receptor properties of SLC1A5 for retroviruses." Can the structure help rationalize this previous report?

The authors write: "Similarly, Arg376 of the transport domain has a potential to form a salt bridge with the Glu264 of scaffold domain in the outward-facing structure (Figure 3B-C). Both Arg376 and Glu264 are highly conserved among the SLC1 family (Figure 1—figure supplement 6)." What are the implications of these observations?

In addition, please address the following issues in the manuscript and/or in responses to the reviewers:

1) Is it known if the Fabs used to aid particle alignment influence transport?

2) The characterization of the transport properties of ASCT2 are based on a single concentration of substrate, which generates a signal to noise ratio of 2. This level of analysis does not instill sufficient confidence in the functional properties of the transporter. It would be preferable to have a measure of substrate affinity, and also demonstration of the exchange properties of the transporter, which would reflect previous studies measuring the functional properties of ASCT2.

3) How significant are the deviations form 3-fold symmetry between protomers? From the superpositions it looks like the deviations are truly minute. Is higher resolution obtained if 3C symmetry is imposed during 3D alignment and refinement? If yes, improved resolution may help analyzing the status of Na^+^ binding sites.

4) Have the authors attempted to perform symmetry expansion to detect whether there is any heterogeneity in the protomers? Such heterogeneity might be masked even if C1 is used for processing.

5) The HDX-MS data is analyzed only very superficially, and as it stands, the level of detail provided does not allow a thorough analysis of the data. On the face of it, these data also do not appear to add significant understanding to the physiological mechanism nor the drug targeting of ASCT2. We recommend removing this section and publishing elsewhere.

[Editors' note: further revisions were requested prior to acceptance, as described below.]

Thank you for resubmitting your work entitled "Cryo-EM structures of the human glutamine transporter SLC1A5 (ASCT2) in the outward-facing conformation" for further consideration at *eLife*. Your revised article has been favorably evaluated by Olga Boudker as the Senior Editor, and a Reviewing Editor.

The manuscript has been improved but there is one remaining issue that needs to be addressed before acceptance, as outlined below:

- Since it appears that little uptake activity is seen (over the background) in either cells or proteoliposomes, please move Figure 1A to the supplementary information and state in the main text explicitly that the observed transport activity was low, compared to the background. Please also use color to distinguish the points in that figure.

---

## [Author Response]

Summary:The neutral amino acid transporter ASCT2 is a member of the SLC1 family of transporters and a target of cancer therapeutics. The manuscript by Yu et al. presents the determination of a structure of ASCT2 in an outward-facing conformation, complementing a previously published structure of an inward facing structure of ASCT2, and providing details of the substrate binding site, which should prove invaluable for the design of drugs to inhibit the transporter.The reviewers appreciate the high quality of the structural work and general importance of the structure, but note that it requires a more direct connection to physiology, e.g. the differences between exchangers and transporters, and/or to drug design. To focus the framing of the article we therefore strongly suggest that you rewrite the article as a Short Report, focus on the substrate and putative CHS binding sites, and exclude the HDX data as explained below.

We would like to thank the reviewers for their valuable time and careful and critical evaluation of our manuscript. We have addressed each of their concerns to the best of our ability as described below. We have rewritten the article as a more concise Short Report that focuses on the substrate and CHS binding sites and we have removed the HDX data as requested by the editor.

Essential revisions:The following are the points that should be considered when improving the physiological context of the structure in the revised manuscript:1) Based on the structures, is it clear why ASCT2 and ASCT1 have different substrate specificity?

Careful comparison of the sequences of ASCT1 and ASCT2 around the substrate binding site show that the most significant difference is at position 467 – ASCT1 has a threonine at this position while ASCT2 has a cysteine. The side chain sulfhydryl group of Cys 467 in ASCT2 appears to form a hydrogen bond with the side chain of glutamine in our structure. Since the structure of ASCT1 is unknown, we attempted to generate a model for ASCT1 by replacing Cys with a threonine. Unfortunately, none of the three rotamers available for threonine can be satisfactorily accommodated because of steric clashes with backbone carbonyl atoms. This suggests that ASCT1 must have a slightly different local conformation around the substrate binding site. Regardless of the local conformation that ASCT1 adopts in this region, the side chain hydroxyl group of Thr459 (analogous to Cys467 in ASCT2) is unlikely to form a productive hydrogen bonding interaction with the glutamine substrate and this could explain why the former transports only fewer amino acids. We have incorporated this description into our revised manuscript.

2) From the drug design perspective please provide more specific comments on the potential applications of the structure for the development of drugs.

SLC1A5 is a high value target in the pharmaceutical industry for the development of drugs for immunomodulatory drugs. The availability of the three-dimensional structure of SLC1A5 in its outward facing conformation open several paths forward for the development of drugs. (1) The structure could be used for identification of chemical leads by virtual screening methods. (2) Secondly, generating additional structures with lead molecules based on substrate-analogs could provide a strong structure-based drug design platform for medicinal chemistry optimization of compound potencies. (3) One important aspect of drug discovery is the need to engineer selectivity of lead compounds for the specific target in a given subfamily. This is especially true for modulators of SLC1A5 where dialing out off-target activities against other closely-related sub-family members is highly desirable. Historically, this has been an area of strength for structure-based design. (4) Structures have also been incredibly useful for optimization of ADME properties of compounds which is a critical element in the overall drug discovery process. (5) We believe the most relevant conformation to target for SLC1A5 inhibitors is the outward-facing conformation that we have described in our current work as that is the most relevant conformation for ligands from extracellular space. We have revised the manuscript to incorporate the above section.

3) It is not clear whether the protein imaged without L-Gln was in high Na^+^ concentration. If yes, then the structure likely represents a Na^+^-bound transporter, which should be evident from the arrangement of the Na^+^-binding residues. If not, or if the affinity for Na^+^ is low, then one would expect to find disrupted Na^+^ binding sites, which again should be visible in the structure.

We agree this is an important question to address. All protein samples in this work were imaged in presence of 100 mM NaCl. Thus, we do believe our apo structure represents Na-bound state of the transporter. We have revised the Materials and methods section to clearly state this. Given the moderate resolution of the structure and the lack of density for sodium and nearby side chains, we are unable to make a categorical statement regarding this. Future structures of ASCTs at different sodium concentrations and at higher resolution could help to clarify the precise role sodium plays in conformational switching.

4) ASCTs are obligate Na^+^-dependent exchangers, while EAATs are concentrative pumps. Could it be rationalized from the structures, why ASCT2 cannot return to the outward-facing states without the bound amino acid and Na^+^ ions?

This is also a very interesting question, but difficult to assess due to the fact that sodium ion is not visible in our structure and we do not have a structure in its absence. It is highly likely that significant conformational changes accompanies binding of sodium ions. Based on our structures in the presence and absence of glutamine, we do know that the presence of substrate does not impede the movement of HP2 loop. It is possible that higher resolution structures with and without both Na^+^ and Gln bound may shed light on this mechanism.

5) Mutations have been made in GltPh mimicking ASCT2 (including R to C mutation), but when the structure was solved with Gln, the HP2 loop was still open. Can it be rationalized from the current structure what is the difference between the binding pockets and/or HP2 between ACST2 and GltPh that allows larger substrates (even when R is mutated to C)?

We believe the editor is referring to the work of Scopelliti et al., 2018, which describes the structures of GltPH (R397C) bound to serine, cysteine or benzylCys. However, there is no Gln-bound structure of GltPh (WT or R397C mutant) that has been reported. In the Cys-bound GltPh-R397C structure, the HP2 loop adopts a “closed” conformation. Similarly, the HP2 loop is closed in one of the protomers in the Ser-bound GltPh structure, However, electron density suggests open conformations for the other two protomers but they were not modeled due to limited resolution. Therefore, it would be difficult to assess/compare structures of ASCT2 and GltPh with bound glutamine or comment on the precise conformation of the HP2 loop.

6) The manuscript methodically indexes many structural features including protomer interactions, helix orientations, key amino acid positions, glycosylation sites, glutamine and CHS binding sites, etc. However, this style is primarily descriptive, so if the authors intend to retain these details, they should to be enhanced with a clearer narrative explaining the significance of the structural findings. Below are three examples.

We thank reviewers for pointing out the description of structural features. Some of them have been streamlined or removed to fit the “short report” style and focus on the significance of the structural findings.

The authors point out that "based on sequence alignment, both SLC1A4 and SLC1A5 have the longest ECL2a region compared to other SLC1 family members including prokaryotic homologues." What is the significance of this?

As there is no significance, we have removed the sentence.

Similarly, the authors write: "Importantly, the ECL2b was reported to play a critical role in determining the receptor properties of SLC1A5 for retroviruses." Can the structure help rationalize this previous report?

As the structure doesn’t rationalize, we have removed.

The authors write: "Similarly, Arg376 of the transport domain has a potential to form a salt bridge with the Glu264 of scaffold domain in the outward-facing structure (Figure 3B-C). Both Arg376 and Glu264 are highly conserved among the SLC1 family (Figure 1—figure supplement 6)." What are the implications of these observations?

In the revised “Short Report” version, we removed the discussion.

In addition, please address the following issues in the manuscript and/or in responses to the reviewers:1) Is it known if the Fabs used to aid particle alignment influence transport?

The biochemical role of the KM4012 for transport activity is currently unknown. It is reported that KM4012 antibody was isolated through a cell-based screen and inhibited glutamine-dependent cancer cell growth (Suzuki et al). The antibodies suppressed glutamine-dependent growth of WiDr colorectal cancer cells, which require glutamine in the culture medium to grow.

In this revised version we added a sentence “It is reported that KM4012 antibody isolated through a cell based screen inhibited glutamine-dependent cancer cell growth. (Suzuki et al)” for further clarification.

2) The characterization of the transport properties of ASCT2 are based on a single concentration of substrate, which generates a signal to noise ratio of 2. This level of analysis does not instill sufficient confidence in the functional properties of the transporter. It would be preferable to have a measure of substrate affinity, and also demonstration of the exchange properties of the transporter, which would reflect previous studies measuring the functional properties of ASCT2.

Thank you for pointing out the weak signal to noise ratio. The level of glutamine transport measured with our SLC1A5 PL assay system was ~6 times lower than the level reported by Pingatore et al. and by Graraeva et al. While Pingatore et al. reported that SLC1A5 in their proteoliposomes (PL) was 100% in the correct orientation (i.e. an orientation needed to measure glutamine uptake PL) only 20-25% of the SLC1A5 in our PL was in that same correct orientation (determined by the availability of an affinity tag in relation to PL surface (data not shown). This could be due to differences in the expression systems that were used by the two different labs. Both Pingatore et al. and Graraeva et al. used a yeast expression system, but our expression system used the mammalian HEK293 cell line. These differences could account for weaker signal we observed in our assay.

In lieu of further PL experiments, we demonstrate that when expressed in cells, the affinity-tagged SLC1A5 used in this study produced a level of glutamine uptake that was comparable to untagged WT SLC1A5, thus suggesting that the presence of the affinity-tags did not alter the biological activity of SLC1A5. To make it more evident that the glutamine transport activity of our tag-version of SLC1A5 is comparable to the wild type protein, we tested a broad range of glutamine concentration and both wild type and the tag version of SLC1A5 showed similar activity. This is depicted in the revised Figure 1A.

We have also explored biophysical studies using SPR but were unsuccessful due to technical challenges with the detergent solubilized protein reagent.

3) How significant are the deviations form 3-fold symmetry between protomers? From the superpositions it looks like the deviations are truly minute. Is higher resolution obtained if 3C symmetry is imposed during 3D alignment and refinement? If yes, improved resolution may help analyzing the status of Na^+^ binding sites.

From the Supplementary Figure 7 (initial submission) on the superposition of each protomer, the results revealed that three protomers from the outward-facing SLC1A5 trimers share similar global fold but are not positioned symmetrically in space. Even though the deviations are truly minute, the fab-bound SLC1A5 trimers adopt the pseudo C3 symmetry and imposing C3 operation during the refinement did not improve the resolution further. The reason could be the presence of the Fab disrupts the low-resolution pseudosymmetry. We further experimented with Fab signals subtraction followed by refinements with/without C3 symmetry operation and the efforts did not improve. the resolution for the SLC1A5-cKM4012 (Fab) structure or the SLC1A5-cKM4012 (Fab)_Gln complex. In the revised “Short Report” version, we removed this figure.

4) Have the authors attempted to perform symmetry expansion to detect whether there is any heterogeneity in the protomers? Such heterogeneity might be masked even if C1 is used for processing.

Since the structures were reconstructed without any symmetry operations, no symmetry expansion was performed. From the 3D classification and the final reconstruction results, only the outward-facing state was observed. Minor heterogeneities were observed at the loop regions among the protomers.

5) The HDX-MS data is analyzed only very superficially, and as it stands, the level of detail provided does not allow a thorough analysis of the data. On the face of it, these data also do not appear to add significant understanding to the physiological mechanism nor the drug targeting of ASCT2. We recommend removing this section and publishing elsewhere.

We have removed the HDX section.

[Editors' note: further revisions were requested prior to acceptance, as described below.]

The manuscript has been improved but there is one remaining issue that needs to be addressed before acceptance, as outlined below:- Since it appears that little uptake activity is seen (over the background) in either cells or proteoliposomes, please move Figure 1A to the supplementary information and state in the main text explicitly that the observed transport activity was low, compared to the background. Please also use color to distinguish the points in that figure.

We appreciate the referee’s concern and comment. We have moved “Figure 1A” to “Figure 1—figure supplement 1A”. The figure was revised in color to distinguish the points. We have also explicitly stated in the main text to read “Our full-length SLC1A5 construct showed high expression and stability. (Figure 1—figure supplement 1). The presence of affinity tags used for purification did not affect the sodium-dependent glutamine uptake when HAP1 SLC1A5 knock-out cells were transiently transfected with full-length SLC1A5 but the observed transport activity was low, compared to the background.” in the revised manuscript.